



# Validation of CAMS AOD using AERONET Data and Trend Analysis at Four Locations in the Indo-Gangetic Basin

Amit Misra[1], Sachchida Tripathi[1], Harjinder Sembhi[2], and Hartmut Boesch[2,3]

[1]Department of Civil Engineering, Indian Institute of Technology Kanpur, India
[2]Earth Observation Science, Department of Physics and Astronomy, University of Leicester, UK
[3]National Centre for Earth Observation, University of Leicester, UK

**Correspondence:** Amit Misra (sri.amit.misra@gmail.com)

**Abstract.** In this work we have validated Copernicus Aerosol Monitoring Service (CAMS) derived aerosol optical depth (AOD) at four locations (Kanpur, Gandhi College, Jaipur and Lahore) in the Indo-Gangetic Basin and used it to study the aerosol climatology and trend in AOD at these locations. Lahore and Kanpur are urban and industrial sites with agricultural activity in the neighbouring regions. Gandhi College is in a rural agricultural area, whereas Jaipur is a desert dust source area. Aerosol climatology at the four sites are examined with MODIS-derived NDVI and ESA-CCI derived soil moisture data. CAMS-derived AOD for black carbon, sulphate, dust, sea salt and organic matter at the four sites are studied and discussed. It is observed that sulphate AOD has the largest influence on the total aerosol climatology. Contribution from dust and sea salt aerosols is observed only during pre-monsoon and monsoon seasons, whereas non-zero AOD is observed for organic matter, black carbon and sulphate aerosols throughout the year at all sites. Comparison of CAMS AOD with AERONET AOD shows better correlation when aerosol climatology is dominated by coarse particles as compared to when it is dominated by fine particles (e.g., at Kanpur, $R^2_{pre-monsoon} = 0.63$ and $R^2_{winter} = 0.36$). Trend analysis shows largest increase in organic matter (e.g., $0.305 \pm 0.021$ per year at Kanpur) and least in sea salt aerosols (e.g., $0.008 \pm 0.001$ per year at Kanpur).





## 1 Introduction

Aerosols are important in the earth climate system due to their interaction with solar and terrestrial radiations, thus affecting the earth radiation budget directly and indirectly. Particles smaller than 1 micron can enter respiratory tracts thereby causing lung cancer (Seaton et al., 1995; Pope-III and Dockery, 2006; Akimoto, 2003; Kampa and Castanas, 2008). Particles smaller than 2.5 microns have significant role in visibility reduction (Han et al., 2012). According to the latest report by the World Health Organisation (Ambient Air Pollution: A global assessment of exposure and burden of disease, *World Health Organisation*, 2016), 9 out of the 10 most polluted cities of the world are located in India within the Indo-Gangetic Basin (IGB). This is the region which is primarily agricultural land, well-irrigated by several major rivers like Sindhu, Ganga, Yamuna, Sutlej and their tributaries. In addition, several centres of high industrial activity are also located in the IGB like Kanpur, Delhi, Lahore, and Ludhiana (Gautam et al., 2010). Hence the aerosol characteristics of this region are complex, having contribution from both agricultural as well as industrial sources. Note that this classification of aerosol sources into agricultural and industrial sources is broad and each of these in turn is quite complex. Thus, agriculture may generate aerosols from dry land left barren after harvest (Ginoux et al., 2012), as well as carbon particles from agricultural waste burning (Rajput et al., 2014). Similarly, though industrial activity implies emissions from industries and factories, it leads to economic development and urbanization, which in turn increases traffic and vehicular emission (Briggs et al., 2000; Gualtieri and Tartaglia, 1998).

For some particular case studies, selected sites in the IGB region are categorised as being agricultural or urban. For example, Delhi is primarily urban whereas Gandhi College located in eastern Uttar Pradesh is agriculture-dominated rural site. However, in general, such demarcation is not easy and both sources of aerosols are present in varying degree. Thus, Jaipur is prone to dust aerosols as well as urban pollution. In addition to transported and local dust, Kanpur is also prone to industrial and vehicular pollution. Further, being downwind of major centres of economic activity like Gurgaon, Delhi, Chandigarh and Agra, the urban pollution here is exacerbated by transported plumes. Although local emission and transport of aerosols seem



to be the primary factor governing the climatology of any region, meteorology also plays an additional significant role. Meteorology aids in build up of aerosols in a particular region and also leads to transport of aerosols from or into a region. Thus, meteorology defines the seasonal variation observed in the aerosol climatology at any site.

Several studies have examined the aerosol types and climatology at individual locations in
the IGB as well as made comparison among different sites. Studies are based on in-situ measurements as well as ground- and space-based remote sensing observations. Dumka et al. (2014) have reported a latitudinal gradient in the observed aerosol properties over IGB. Eck et al. (2010) and Giles et al. (2011) have studied aerosol optical depth (AOD) vs absorption aerosol optical depth (AAOD) and angstrom exponent (AE), and classified aerosol types into different cate-
gories. Srivastava et al. (2011) have shown that aerosol climatology in eastern IGB is dominated by fine mode particles as compared to central IGB. Based on prior knowledge about geographical and geological characteristics of these sites, the observed aerosol climatology is interpreted in terms of meteorological or land surface properties.

In this paper, we first present aerosol climatology at four representative locations in the IGB,
viz., Kanpur, Gandhi College, Lahore and Jaipur, from ground-based remote sensing data. We demonstrate how the observed variation in aerosol optical properties can be related to climate and meteorological factors. These locations are situated across the expanse of IGB region with different local meteorology and geographical characteristics. Thus, Jaipur is primarily a dust source region, Kanpur and Lahore are urban centres surrounded by agricultural areas, Gandhi
College is primarily a rural area dominated by agricultural activity. Also, the monsoon pattern is different at these sites, e.g., monsoon reaches Lahore last and recedes from there first. More details about the sites are given in section 2.

We have examined land surface characterisics using soil moisture data from the European Space Agency Climate Change Initiative (ESA-CCI) and normalized difference vegetation in-
dex (NDVI) from Moderate Resolution Imaging Spectroradiometer (MODIS) to understand the effect of earth surface on the prevailing aerosol climatology at the selected sites. We used ground-based AOD data to validate Copernicus Aerosol Monitoring Service (CAMS) derived AOD data, and then used the latter to derive climatology for individual aerosol components and





study their trend at the four sites. The CAMS AOD product is described in section 3, and details

of trend analysis are given in section 4.4.

   In the next section, we describe the four sites and their peculiar characteristics relevant to the

present study. In section 3, details of various data products, source of data and data analysis are

given. In section 4, we explain the results from our study.

## 2   Site description

Kanpur is located in central IGB on the bank of the river Ganga and is a major centre of eco-

nomic activity. The Aerosol Robotic Network (AERONET) site is situated in the campus of

the Indian Institute of Technology Kanpur, at a distance of about 25 km from the city cen-

tre (Dey et al., 2005). Kanpur witnesses heavy storms during pre-monsoon season (Dey et al.,

2004; Misra et al., 2014) and fog during winter (Tripathi et al., 2006; Kaskaoutis et al., 2012).

Temperature at Kanpur varies from 9 °C during winter (Tripathi et al., 2006) to more than 40

°C during summer (Misra et al., 2014). Relative humidity varies from 9% during pre-monsoon

season to more than 90% during monsoon season (Misra et al., 2014).

   Gandhi College is located about 466 km in east of Kanpur and is a predominantly rural site

(Varpe et al., 2018). Gandhi College is mainly agricultural area, well-irrigated by ground water

and monsoon rainfall. The main crops grown in this area are wheat, paddy, mustard and pulses.

The aerosol climatology at Gandhi College is dominated by fine mode particles as compared

to stations in the central IGB (Srivastava et al., 2011). Being an agricultural area, effect of land

surface parameters on aerosol climatology is likely to be more noticeable at Gandhi College as

compared to other sites.

Jaipur is situated in the western Indian state of Rajasthan and is a desert dust source region

(Verma et al., 2013). During pre-monsoon season, dust from this region is transported to eastern

parts of IGB. Jaipur is situated in south-west direction to Delhi. The city has several small-scale

industries. Variation in temperature is large and it can vary from a minimum of 3 °C during

winter to 48 °C during summer. Relative humidity varies from 20% to 80%.





Lahore is a semi-arid site in north-west IGB. It lies in the Punjab province of Pakistan and

is situated near its border with India. Lahore is a major industrial hub of Pakistan with several

manufacturing industries located there. The temperature at Lahore varies from 7 $^{o}$C in winter

to 39 $^{o}$C during summer (Alam et al., 2012). Relative humidity varies from 34% to 68%. Major

sources of aerosols at Lahore are dust, industrial and vehicular emission, and biomass burning

(Alam et al., 2012). Aerosol optical depth (AOD) at Lahore varies between 0.57 and 0.76 and

angstrom exponent (AE) between 0.39 and 1.22 (Alam et al., 2012) which shows presence

of a variety of particle sizes during different seasons. The land of Lahore is very fertile and

well-irrigated by canals. Among the sites considered in this study, Lahore is the westernmost

and northernmost site. As noted previously, monsoon reaches Lahore around 15 July and starts

receding by 1 September. Thus, monsoon reaches Lahore last and recedes first. The AERONET

station is situated within the campus of Institute of Space Technology Lahore. It should be noted

that although Lahore district has large area of agricultural land, the city Lahore itself is a major

urban site having many industries and other centres of high economic activity (Ali et al., 2014;

Alam et al., 2012). Due to high moisture and extremely low tempteratures, this site witnesses

heavy fog during winter (Yasmeen et al., 2012). In addition, agricultural waste burning results

in release of large concentration of fine mode particles (Tariq et al., 2015).

## 3   Data products and methodology

In this study, we have examined aerosol optical properties only. Ground based data are obtained

from the AERONET sunphotometer network (Holben et al., 1998). AERONET sunphotometer

measures direct solar radiation in 7 wavelength channels and retrieves AOD and AE using Beer

Lambert Law (Holben et al., 1998). The period of study is taken as 2006 to 2015 in order to

maintain consistency between different sites. Version 2 monthly averaged data products are used

in this study. For Kanpur and Gandhi College, data is taken from April 2006 to November 2015,

for Lahore data is taken from December 2006 to April 2015, and for Jaipur data is taken from

from April 2009 to December 2015.



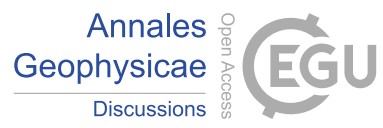

1 km resolution global vegetation indices data from MODIS (MOD13A3) Collection 6 for the period January 2006 to December 2015 is used in this study. MODIS retrieves vegetation indices from the corrected reflectances in the red, blue and near–IR wavelengths. We have considered only the normalized difference vegetation index (NDVI), which is used to relate reflectance data with crop parameters. Value of NDVI varies from -1 to +1, and is defined as:

$$\text{NDVI} = \frac{R_{IR} - R_{Red}}{R_{IR} + R_{Red}}$$

where $R_{IR}$ is the reflectance at infra-red wavelength and $R_{Red}$ is the reflectance at red wavelength (George Joseph, *Fundamentals of Remote Sensing*, Universities Press, 2005).

Version 3.2 of the soil moisture data from the European Space Agency Climate Change Initiative (ESA CCI) is used in this study. The combined product at 0.25 degrees resolution is used (Gillies et al., 1997). This product is retrieved from a combination of active and passive remote sensing data from a variety of sources.

We used global reanalysis of different aerosol types which were extracted from the Copernicus Atmosphere Monitoring Service (CAMS) atmospheric composition (AC) dataset produced by the European Centre for Medium-Range Weather Forecasting (ECMWF)(Flemming et al., 2017). The aerosol reanalysis consists of global three-dimensional fields calculated on 80 km (T255) and at time-steps of 3 hourly fields with 1 hourly fields calculated for surface forecasts. Sources of aerosol in our analysis include, black carbon (BC), sulphates (S), sea salt (SS), dust (D) and organic matter (OM) as well as estimates of total AOD.

CAMSRA aerosols are calculated with the ECMWF Integrated Forecast System (IFS) aerosol and chemistry module which is a hybrid scheme consisting of 12 tracers. Different aerosol types are treated as externally mixed, separate particles. Advection, convection and diffusion are handled by meteorological component of IFS which includes dry and wet deposition. Total AOD analysis fields are corrected by real-time assimilation of satellite AOD fields which include the Collection 6 MODIS AOD fields and those from the ESA-CCI Advanced Along Track Scanning Radiometer AOD product (2003 — 2012). Inness et al. (2019) presented an assessment





of performance average of 230 AERONET V3 Level 2.0 sites Southeast Asia which find that

CAMSRA AOD has a bias and SD of 0.013±0.176.

## 4  Results and discussion

### 4.1  Influence of surface properties on aerosol climatology

Role of land surface on aerosol properties is clearly seen at Kanpur during pre-monsoon and monsoon seasons (figure 1). During pre-monsoon season, soil moisture is low and vegetation

is also less, both these factors lead to release of more dust particles into air. Hence, the high AOD in this season is mainly due to dust aerosols, which is also reflected in the low value of AE. With the onset of monsoon, a drop in AOD and an increase in AE is observed. First, due to rainfall, particles in atmosphere are washed out. Secondly, in the context of this study, the increased vegetation holds the soil together. This factor, along with higher soil moisture, reduce

the amount of soil-generated aerosols. Another peculiar feature is seen during winter season when vegetation is less and soil is dry. In this season, AOD as well as AE increase. The high value of AOD in this season is mainly due to finer particles, as is reflected in the high value of AE.

Overall, the variation in AOD, AE, NDVI and soil moisture at Gandhi College are similar to

that at Kanpur, with magnitudes different. As mentioned earlier, Gandhi College is agricultural rural area, hence the value of NDVI is higher than at Kanpur. AE during pre-monsoon season is higher than at Kanpur, showing that during pre-monsoon season, particle size is smaller at Gandhi College than at Kanpur.

At Jaipur, the pattern of AOD and AE is very different from that at Kanpur and Gandhi

College. Unlike Kanpur and Gandhi College, where AOD reached its maximum value during May-June, at Jaipur AOD reached maximum value in July. However, similar to Kanpur and Gandhi College, at Jaipur also AE had low value in May-June. Thus, there is a one month difference between maximum of AOD and minimum of AE. As mentioned previously, in case of Kanpur and Gandhi College, these two points coincided and led to the conclusion that soil-

generated aerosols are the primary contributors to the high AOD in pre-monsoon season. The





value of AE in May-June at Jaipur is even lower than all other sites. Thus, not only the aerosol climatology during pre-monsoon season at Jaipur is dominated by coarse particles, the fraction of these particles is higher than that at other sites. However, as mentioned in the last paragraph, AOD does not attain its peak value before July, when AE has already started increasing. This

means that the pre-monsoon soil-generated dust aerosol loading is not the reason behind the peak in AOD during July. One possibility is that during July, the wind direction is easterly (Misra et al., 2014), due to which particles from the highly polluted region to the east of Jaipur flow into Jaipur, thereby increasing the AOD. This seems to be a plausible reason considering that the NDVI value during monsoon season is similar to that at Kanpur. Though Jaipur is in a

dust source region, the site is vegetated as reflected in the NDVI value which is similar to that at Kanpur. NDVI values during February to April is much lower than Kanpur and Gandhi College both, which shows that surface characteristics of Jaipur are different from those sites.

AOD climatology at Lahore is similar to that at Jaipur. The minimum value of AE is attained during June and AOD reaches maximum value during July. Hence, similar to Jaipur,

soil-generated dust aerosols cannot be the reason behind the high value of AOD in July. Considering the fact that during July, wind direction is easterly and highly polluted regions including Delhi lie to the east of Lahore, it seems plausible that particles from these locations flow into Lahore. The aerosol climatology at Lahore is closer to Jaipur in characteristics than Kanpur and Gandhi College. However, the maximum AOD at Lahore, attained in July, is even higher than

Jaipur. Also, the increase in AOD from pre-monsoon to monsoon season at Lahore is very sharp as compared to other sites. AE during June is lower than Gandhi College but higher than Jaipur. Similar to Kanpur, aerosols at Lahore are more of a polluted dust in nature. NDVI characteristics at Lahore are similar to Kanpur and Gandhi College, but its value is lower than these two sites. Not much variation is noted in soil moisture at Lahore, except an increase in July.

**4.2   Correlation of AERONET and CAMS AOD**

We performed a correlation analysis of CAMS total AOD versus AERONET observed AOD for the whole study period (denoted by 'All') as well as for different seasons. The objectives of these inter-comparisons were to: a) first assess the ability of CAMS to capture total AOD



variations observed in the four AERONET stations and b) understand which aerosol type (as

simulated by CAMS) dominates any biases or differences observed in CAMS and AERONET

differences. By performing longer-term trend analysis, we will evaluate whether behaviour of

CAMS aerosol types is expected over each AERONET station with respect to the dominance of

surface properties discussed in section 4.1. The mean bias, standard deviation and $R^2$ of corre-

lation are given in table 1 for different study locations. The seasons are pre-monsoon (March –

May), monsoon (June – August), post-monsoon (September – November) and winter (Decem-

ber – February) (Choudhry et al., 2012). In addition, correlation result for climatological mean

is also given in the table (denoted by 'Clima').

1. For all stations, the correlation coefficient is better for climatological mean than with full

   time series (e.g., in the case of Kanpur, $R^2 = 0.37$ for full time series and $R^2 = 0.49$ for

climatological mean, and similarly for other sites). This is in agreement with Inness et al.

   (2019) who recommended using CAMS data for climatological studies.

2. Except Lahore, the correlation is best in pre-monsoon season (dominated by coarse mode

   particles), and poorest during winter (dominated by fine particles) including Lahore. For

   example, in the case of Kanpur, $R^2_{pre-monsoon} = 0.63$ and $R^2_{winter} = 0.36$, and similarly for

Gandhi College and Jaipur. Thus, CAMS performs better when climatology is dominated

   by coarse mode particles, and poor when fine mode particles are dominant.

3. The correlation during monsoon is closer to that during post-monsoon as compared to

   other seasons. For example, for Kanpur, $R^2_{monsoon}=0.47$ and $R^2_{post-monsoon}=0.48$, and sim-

   ilarly for other sites.

4. The observation that the correlation at Lahore is best during monsoon and post-monsoon

   is likely due to different classification of the annual cycle into seasons. At Lahore, mon-

   soon arrives late, i.e., during July (as compared to June at other sites) and leaves in

   September. Thus while May is dominated by coarse particles at other sites, at Lahore

   coarse particles are dominant during June.





5. Correlation at Jaipur (dust dominated) is better than at Kanpur and Gandhi College during all seasons. For example, during pre-monsoon season, $R^2_{Kanpur} = 0.63$, $R^2_{GandhiCollege} = 0.68$, $R^2_{Jaipur} = 0.81$, and similarly for other seasons. Also, except monsoon, the mean bias is negative, implying that CAMS derived AOD is lower than AERONET AOD.

    6. In general, the standard deviation is lower at Jaipur than other sites, implying that the
variability in correlation across seasons is lower at Jaipur than other sites.

Figure 2 shows the mean bias ($AOD_{CAMS} - AOD_{AERONET}$) in climatological mean AOD at the four sites. It is seen that at Kanpur and Gandhi College the bias is highest during June to September. This is the only period when the bias is positive (> zero), i.e., CAMS value is higher than AERONET. At Lahore the bias is large and positive during February and again during June
and July. Even though the climatological mean has better correlation with AERONET AOD as compared to whole time series, this figure shows the variability in the bias, which is seen to be different for different seasons.

### 4.3    Climatology of different aerosol types using CAMS data

Figure 3 shows the climatology of different aerosol species and total AOD based on CAMS data.
This analysis is helpful in understanding the relative concentration of aerosol species during different seasons and their contribution to overall aerosol climatology of the study location. Although major aerosol types are same at the four sites, their relative concentration could be different.

The most prominent feature in dust climatology is that the pattern of variation in dust AOD
is nearly similar at all sites, showing high dust AOD during pre-monsoon season (ranging from ~0.2 at Gandhi College to ~0.4 at Lahore) and nearly zero during winter at all sites. This is an improvement in CAMS data from previous version (not shown) which showed non-negligible dust AOD during winter months also. Thus, the updated CAMS AOD is reasonable considering the known aerosol climatology from in-site observations. Another feature is that for Kanpur
and Gandhi College, maximum AOD is observed during May and June, whereas at Lahore, AOD reaches maximum value during June and July. This is due to the one month delay in onset





of monsoon at Lahore as compared to that at Kanpur and Gandhi College. Thus pre-monsoon season consists of different months at these sites. At Jaipur, high dust AOD is observed during May to July. During January to May, dust AOD at all sites are nearly similar except Gandhi

College, where AOD is low during May. During June to September, Lahore has maximum AOD among all sites. Gandhi College shows lowest dust AOD during May to September.

The variation in organic matter optical depth at Kanpur, Gandhi College and Lahore are in general similar, with difference in absolute magnitude, except during monsoon and winter. However, OM optical depth at Jaipur is lowest among all sites during all seasons. Another feature

to be noted is that in contrast to dust, OM AOD is non-zero during all seasons, and also its magnitude is significant. Largest variation in OM AOD is observed at Kanpur and Gandhi College between winter and pre-monsoon seasons, and also their values are similar throughout the year, except December. Although OM AOD is low during pre-monsoon season as compared to winter, its magnitude is still comparable to dust AOD during this season.

Sea salt AOD is non-zero only during pre-monsoon and monsoon seasons. During other seasons, aerosol climatology is not influenced by sea salt aerosols. During pre-monsoon and monsoon seasons, wind from south-west brings in sea salt aerosols to IGB. This is reflected in the magnitude of sea salt AOD at different sites during these seasons. While its value is same at Kanpur and Gandhi College ($\sim$0.025), it is maximum at Jaipur and lowest at Lahore. In other words,

contribution of sea salt aerosols decreases with increase in latitude. Most of these aerosols are transported from Arabian Sea. Sea salt AOD is much lower as compared to dust and organic matter AODs.

Black carbon AOD is also less as compared to dust and organic matter. Its value lies in the range 0.01-0.03. Variation in BC AOD is very less; its value is in general low during monsoon

season and maximum in winter months. However, at Lahore, high BC AOD is observed during pre-monsoon season also. BC AOD at Jaipur is lowest among all sites during all seasons.

Sulphate AOD is lowest at Jaipur, and in general similar at the other three sites. In general, sulphate AOD at Jaipur and Lahore show similar pattern, except October and November. Note that sulphate aerosol climatology has the largest influence on the total aerosol climatology. At

all sites, sulphate AOD is maximum during monsoon season because of increase in conversion



rate of sulphur dioxide to sulphate aerosol under high humidity conditions. However, sulphate AOD does not decrease much after monsoon; its value is lowest during March to May at all sites.

Overall aerosol climatology (total AOD) is nearly similar at the four sites, with different

magnitudes of AOD. Also, there is a time lag between the period of maximum AOD. At Kanpur and Gandhi College, maximum value of AOD is observed during May and June, whereas that at Lahore and Jaipur is maximum during July. It seems plausible that this maximum value coincides with onset of monsoon.

Among all sites, total AOD at Jaipur is lowest throughout the year, except during July. AOD

at other three sites are nearly similar, except at Lahore during June to October.

It is interesting to note that at Kanpur, Gandhi College and Jaipur, wintertime AOD is also high, either comparable or even higher than that during pre-monsoon season. However, at Lahore, AOD does not increase during winter and its value remains very low as compared to its pre-monsoon value. It shows that wintertime aerosol climatology at Lahore is governed by dif-

ferent phenomenon than that at Kanpur, Gandhi College and Jaipur.

### 4.4 Trend analysis of CAMS AOD

We investigated the long-term and regional influences on different aerosols at these four locations using a time-series from 2003 to 2016 to map the long-term and short term variability, growth rate and trends. Our method used a digital curve-fitting technique (Thoning et al., 1989)

to each aerosol time-series which consists of a function fit that is a combination of a polynomial and annual harmonics. The polynomial fit approximates the shape of the aerosol time-series and a liner least squares regression methods fits the function to the aerosol data with a Numerical Recipes "LFIT" routine in Fortran. The residual between the data and the fit function are calculated (to capture short-term variations) after transforming the data from the time (daily)

domain to the frequency domain with a Fast Fourier Transform (FFT) algorithm which is then multiplied by a low pass digital filter to filter short-term variations/noise (Pickers and Manning, 2015). The filtered data are transformed back to the original time domain using an inverse FFT.





Figure 4 shows the trend curve of AOD at different sites due to individual species and total aerosol. The magnitudes of mean trend and standard deviation for different aerosol types and total aerosol at different sites are given in table 2. Following observations are made in the results of trend analysis.

1. Lowest trend is observed in sea salt aerosols (e.g., 0.008±0.001 at Kanpur), followed by black carbon aerosols (e.g., 0.020±0.001 at Kanpur). Also, the standard deviation is low, implying that the increase is at a nearly constant rate.

2. Largest trend is seen in organic matter (e.g., 0.305±0.021 at Kanpur) followed by sulphate aerosols (e.g., 0.172±0.020 at Kanpur). The value of standard deviation is larger as compared to black carbon and sea salt aerosols. Thus the rate of increase in organic matter and sulphate is higher than other aerosol types.

3. Increase in dust is not uniform across all sites. The trend is higher at Lahore (0.141±0.017) and Jaipur (0.118±0.014), both sites in the vicinity of Thar desert, as compared to Kanpur (0.096±0.013) and Gandhi College (0.063±0.011).

4. Increase in black carbon is similar at all sites, except Jaipur.

5. Among all sites, the trend is lowest at Jaipur for all aerosol types, except dust. Also, the magnitude of AOD for individual aerosol types is lowest at Jaipur as stated earlier.

This illustrates the application of CAMS data, as it is able to identify the aerosol type which has the largest influence on overall trend in AOD.

## 5  Conclusions

AERONET-derived optical properties at Kanpur, Gandhi College, Lahore and Jaipur show that pre-monsoon aerosol loading at all the four sites is primarily due to dust aerosols. At Kanpur and Gandhi College, this results in highest values of AOD, which remains high during monsoon season also, as a result of meteorological factors explained earlier. On the other hand, at Jaipur and Lahore, maximum AOD is reached during monsoon, mainly due to easterly wind direction,

due to which aerosols from neighbouring regions flow into these sites. Kanpur and Gandhi College show high AOD during winter season also, during which AE is also high. This shows that

winter-time aerosol climatology at these sites is primarily due to anthropogenic aerosols. Jaipur and Lahore do not show any such feature. Though AE at these sites is also high implying that anthropogenic aerosols are the primary component, yet AOD is lower than Kanpur and Gandhi College. NDVI at Kanpur, Gandhi College and Lahore shows similar pattern, with different amplitudes. At Jaipur, NDVI is similar to Kanpur in monsoon, but lower than other three sites

during all other seasons.

Correlation analysis of CAMS AOD with AERONET AOD shows that CAMS performs better when climatology is dominated by coarse mode particles as compared to when fine mode particles are dominant (e.g., at Kanpur, $R^2_{\text{pre--monsoon}} = 0.63$ and $R^2_{\text{winter}} = 0.36$). CAMS-derived aerosol climatology shows that sulphate AOD has the largest influence on total aerosol clima-

tology. Contribution from dust and sea salt aerosols is observed only during pre-monsoon and monsoon seasons, whereas non-zero AOD is observed for organic matter, black carbon and sulphate aerosols throughout the year at all sites. Trend analysis using CAMS data shows maximum increase in organic matter (e.g., 0.305±0.021 per year at Kanpur) and lowest increase in sea salt aerosols (e.g., 0.008±0.001 per year at Kanpur).

*Author contributions.* AM did the data analysis, data visualization and manuscript preparation, HS provided CAMS data and trend analysis code, and contributed to manuscript preparation, SNT and HB were involved in project administration and supervision.

*Competing interests.* No competing interests

*Acknowledgements.* This work is funded by the UK-India Education and Research Initiative (UKIERI)

project. We acknowledge the ESA CCI Land Cover project for the land cover data used in this work. We



thank the AERONET team for maintaining the sites and providing the data used in this study. MODIS data

was downloaded from Earthdata Search.





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

**Figure 1.** Variation of aerosol optical depth (AOD), angstrom exponent (AE), normalized difference vegetation index (NDVI) and soil moisture at Kanpur, Gandhi College, Lahore and Jaipur. See text for discussion.





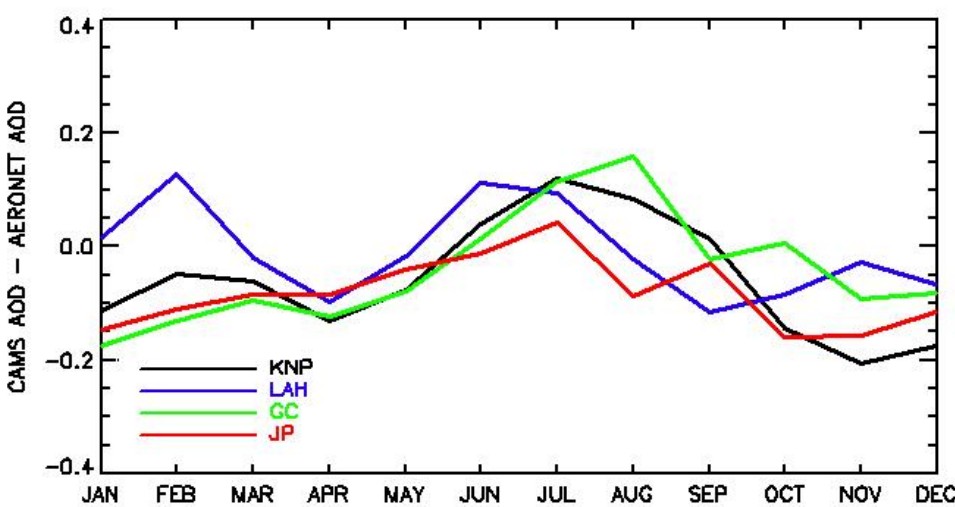

**Figure 2.** Mean bias in CAMS-derived total AOD climatology from AERONET-observed AOD.





**Figure 3.** Climatology of CAMS-derived aerosol optical depth (AOD) at 550 nm for dust, organic matter, sea salt, black carbon, sulphate aerosols and total AOD over Kanpur, Gandhi College, Lahore and Jaipur for January 2006 to December 2016.





**Figure 4.** Trend in CAMS-derived AOD due to dust, black carbon, organic matter, sea salt, sulphate and total aerosol at Kanpur, Lahore, Gandhi College and Jaipur.





| Case | Kanpur | | | Lahore | | | Gandhi College | | | Jaipur | | |
|---|---|---|---|---|---|---|---|---|---|---|---|---|
| | mean bias | stdev | $R^2$ | mean bias | stdev | $R^2$ | mean bias | stdev | $R^2$ | mean bias | stdev | $R^2$ |
| All | 0.011 | 0.134 | 0.37 | 0.047 | 0.134 | 0.55 | 0.028 | 0.152 | 0.39 | -0.034 | 0.080 | 0.73 |
| PreM | -0.052 | 0.086 | 0.63 | -0.017 | 0.104 | 0.29 | -0.060 | 0.069 | 0.68 | -0.062 | 0.040 | 0.81 |
| Mon | 0.132 | 0.099 | 0.47 | 0.113 | 0.090 | 0.73 | 0.132 | 0.169 | 0.51 | 0.027 | 0.094 | 0.56 |
| PostM | -0.039 | 0.128 | 0.48 | -0.019 | 0.092 | 0.79 | 0.049 | 0.127 | 0.47 | -0.057 | 0.067 | 0.59 |
| Win | -0.003 | 0.135 | 0.36 | 0.114 | 0.167 | 0.01 | -0.020 | 0.153 | 0.38 | -0.055 | 0.070 | 0.48 |
| Clima. | -0.058 | 0.103 | 0.49 | -0.008 | 0.082 | 0.78 | -0.042 | 0.100 | 0.58 | -0.081 | 0.062 | 0.80 |

**Table 1.** Mean bias, standard deviation and correlation coefficient of CAMS vs AERONET comparison.





| Case | Kanpur | Lahore | Gandhi College | Jaipur |
|---|---|---|---|---|
| Dust | 0.096±0.013 | 0.141±0.017 | 0.063±0.011 | 0.118±0.014 |
| Organic Matter | 0.305±0.021 | 0.315±0.019 | 0.328±0.022 | 0.152±0.010 |
| Sea Salt | 0.008±0.001 | 0.006±0.001 | 0.009±0.001 | 0.011±0.002 |
| Black Carbon | 0.020±0.001 | 0.022±0.001 | 0.022±0.001 | 0.011±0.001 |
| Sulphate | 0.172±0.020 | 0.185±0.019 | 0.187±0.019 | 0.100±0.009 |
| Total | 0.592±0.038 | 0.594±0.044 | 0.599±0.039 | 0.392±0.026 |

**Table 2.** Mean trend and standard deviation in CAMS-derived AOD for dust, black carbon, organic matter, sea salt, sulphate and total aerosols at Kanpur, Lahore, Gandhi College and Jaipur.