# Peer review of "Validation of CAMS AOD using AERONET Data and Trend Analysis at Four Locations in the Indo-Gangetic Basin"

_Annales Geophysicae, 2020_

## Referee Comment (RC1) · Anonymous Referee #1 · 21 Jul 2020

General Comments:

The manuscript examines multi-year monthly climatology values of total aerosol optical depth (AOD) from CAMS and AERONET centered on four AERONET sites (i.e., Kanpur, Gandhi College, Jaipur, and Lahore) in the Indo-Gangetic Basin (IGB). Further, the manuscript presents CAM total AOD and partitioned into sub-categories (Dust, Organic Matter, Sea Salt, Black Carbon, and Sulphate) to provide climatology and trend analysis. The manuscript suggests better comparison of CAMS to AERONET when dust is the predominant aerosol type.

Major points that need to be addressed before publication:

– Replace Version 2 AERONET data with Version 3 AERONET data; see https://amt.copernicus.org/articles/12/169/2019/ – Provide clarification on matchup between CAMS and AERONET – Provide more description on why CAMS AOD climatology mean does not capture the AERONET AOD climatology mean –Provide more analysis on the trend curve which appears to be lacking key information

Detailed Comments: Line 20: 1 micron radius or diameter? Please state.

Line 34: "from industries and factories." Please be more specific. For example, textile industry, power generation, refining, etc.

Lines 92-94- Kanpur urban area is surrounded by agricultural region so it is also affected by land surface parameters and this is seen in Figure 1.

Line 117: This section "Data products and methodology" lacks information on how the CAMS and AERONET data sets were matched up. The AERONET data are performed during cloud-free periods during the day. Is CAMS taken for the entire day (24-hour period) or matched only when AERONET only measured? If CAMS is taken for the entire day, then this introduces significant bias and uncertainty into the data set comparison.

Line 122: Need to use AERONET Version 3 and not Version 2.

Line 147: Please define "CAMSRA"

Lines 170-173: Gandhi College is further downwind from dust source compared to other sites and thus most dust deposits before reaching the site. As a result, the particle sizes will likely be lower than other sites downwind of the dust (i.e., Jaipur and Kanpur). Please describe this process in the text.

Lines 193-198: Since wind flow is important here, please provide back trajectory analysis.

Lines 207: Indicate month and year of "whole study period"

Line 239-240: CAMS shows large seasonal variation +/-0.1 in total AOD compared to

[Figure]

AERONET mean. Seasonal statistics also indicate large standard deviation >~0.1 in total AOD. Please discuss why CAMS does not identify aerosol variation. The data sets may not be matched up properly as discussed in Line 117 comment?

Line 248: Section 4.3: Discussion in this section evaluates each aerosol subtype; however, the paragraphs in this section do not provide evidence of these aerosol subtypes or references to previous work that discuss the occurrence of these aerosol subtypes. Please provide additional references to previous work.

Lines 301-305: This paragraph is difficult to follow and needs revision.

Line 318: Figure 4: In the Dust Trend Curve plot, how representative is the trend with significant increases in 2009? What is the error in the CAMS seasonal amplitude and does it provide the same magnitude as the difference with CAMS and AERONET?

Line 320: Table 2: The trend values do not indicate increasing, decreasing, or near constant variation with time. Instead, these trend values appear as the mean and standard deviation of the trend curve itself. From a climatological perspective, these values do not hold much value or much more description is necessary to justify them in conjunction with information indicating the change in aerosol loading at these four locations.

Lines 351-353: Why is this the case? For example, does CAMS have better soil moisture parameterization than emission factors?

Line 358: "(e.g., 0.305+/-0.021 per year at Kanpur)" If the organic matter AOD trend is 0.305 per year increase at Kanpur, this magnitude is extremely high and not physical. In fact, this interpretation then makes Table 2 suspect. For Kanpur, total AOD value trend of 0.592 per year is not physical.

---

## Referee Comment (RC2) · Anonymous Referee #2 · 28 Aug 2020

The manuscript deals with validation of CAMS data using AERONET network data from four stations in Indo-Gangetic Basin over ten years. It brings some useful basic comparison of data but the methods are not described in sufficient detail so that the results could be compared/used for some other studies. The presentation may be also improved.

Specific comments: It is not clear what the aerosol climatology should be – annual cycle? Annual mean? Or the 10-years averages?

Also within the MS, it is not clear what time resolution are individual analyses based on – monthly data, annual means? For example, L218 etc.

[Figure]

In the abstract, the level of details varies a lot – for example while it is not clear what data were used for comparison, trend of sea salt in Kanpur is given. . .

What meteorological variables aid to build up aerosol in the region? Please be more specific, is it inversions, pressure field, any wind feature? L46

It is necessary to describe the AERONET data used in the comparison –instruments type and manufacturer, wavelength used, data validation procedure, sampling frequency, data coverage, etc. The measurement location could be also plotted in a figure.

What would be the source of the smaller particles at Gandhi College? L173

Several times through the MS, the source of the particles, air flows etc. are discussed without any analyses to support the claims – could back trajectories clustering be applied, or CPF or PSCF functions calculated?

It is not clear why the monsoon periods were not location-dependent in the comparison? If they were, the results discussion would be easier (L230. . .)

Is it possible to highlight what is new in the section 4.3? It looks only as a description of data without any new analyses?

Any idea what the different phenomenon governing aerosols in Lahore could be? L305

For all the plots, units need to be added, and annual cycle used instead of climatology. The uncertainty is mean $\pm$ st .dev? Not stated. In Fig 1, the monsoon periods could be highlighted in the plots and a different color for NDVI used, as it is not easy to distinguish between NDVI and AE. In Fig. 4/Table 2, was the statistical significance of the trend tested? Some of them seem quite constant.

Technical corrections: - PM does not cause only lung cancer L20 - instead of microns, use of $\mu$m would be preferable? L20 - central IGB is dominated by coarse mode? L56 - paragraph L76 to L78 only repeats the previous text - What climate factors did you

consider? L61 - Would not it be preferable to take the same period for all stations for a better comparability? L123 - One-month shift is not "very different" pattern? - How did you estimate the fraction of coarse particles? L182 - Was there different time span in 4.4. part or is it a typo? L308